# Effect of Exercise Duration on Postprandial Glycaemic and Insulinaemic Responses in Adolescents

**DOI:** 10.3390/nu12030754

**Published:** 2020-03-12

**Authors:** Karah J. Dring, Simon B. Cooper, Ryan A. Williams, John G. Morris, Caroline Sunderland, Mary E. Nevill

**Affiliations:** Sport Health and Performance Enhancement (SHAPE), Sport Science Department, School of Science and Technology, Nottingham Trent University, Nottingham NG11 8NS, UK; karah.dring@ntu.ac.uk (K.J.D.); simon.cooper@ntu.ac.uk (S.B.C.); ryan.williams2013@my.ntu.ac.uk (R.A.W.); john.morris@ntu.ac.uk (J.G.M.)

**Keywords:** high-intensity intermittent exercise, postprandial, glycaemic, insulinaemic, adolescents, duration, frequency

## Abstract

High-intensity intermittent exercise (HIIE) is a potential intervention to manage hyperglycaemia and insulin resistance in adolescents. The aim of this study was to determine the optimum duration of HIIE to reduce postprandial glycaemic and insulinaemic responses in adolescents and the longevity of the response. Thirty-nine participants (12.4 ± 0.4 year) completed a 30- and 60-min exercise trial (Loughborough Intermittent Shuttle Test) and a rested control trial in a randomised crossover design. Capillary blood samples were taken at baseline, immediately and 1-h post-exercise; and 30, 60 and 120 min following a standardised lunch (day one) and a standardised breakfast 24-h post-exercise. Plasma insulin total area under the curve (tAUC) following lunch was lower following 60-min HIIE (21,754 ± 16,861 pmol·L^−1^ × 120 min, *p* = 0.032) and tended to be lower following 30-min HIIE (24,273 ± 16,131 pmol·L^−1^ × 120 min, *p* = 0.080), when compared with the resting condition (26,931 ± 21,634 pmol·L^−1^ × 120 min). Blood glucose concentration was lower 1-h post-exercise following 30-min HIIE (3.6 ± 0.6 mmol·L^−1^) when compared to resting (4.1 ± 0.9 mmol·L^−1^, *p* = 0.001). Blood glucose and plasma insulin concentration did not differ across trials on day two. Shorter bouts of HIIE (30-min), as well as a 60-min bout, reduced the postprandial insulinaemic response to lunch, an ecologically valid marker of insulin sensitivity. As the beneficial effects of HIIE were limited to 3 h post-exercise, adolescents are recommended to engage daily HIIE to enhance metabolic health.

## 1. Introduction 

Since the first diagnosis of type 2 diabetes in adolescents in the UK (in 2000), there has been an exponential increase in the incidence of the condition with ~1400 adolescents currently diagnosed [1]. The increased prevalence of type 2 diabetes in adolescents is of concern given the prognosis of the condition in adulthood worsens with increased exposure to the underlying metabolic risk factors (including hyperglycaemia and insulin resistance) [2]. Thus, preventing the development of metabolic risk factors in adolescents is an important public health concern [3]. Regular participation in physical activity is a recommended, cost-effective therapeutic intervention that has previously been successful in protecting against the development of such metabolic risk factors in young people [4]. As such, government guidelines recommend adolescents participate in a minimum of 60-min moderate-to-vigorous physical activity per day [5], with adverse health trends in adolescents having been attributed to insufficient physical activity levels [6]. The most recent data in the UK suggest that only 18% of young people achieve the recommended 60-min moderate-to-vigorous physical activity per day [7]. Despite the importance of physical activity in enhancing metabolic health, there is no information available relating to the optimum mode, duration or intensity of physical activity to improve metabolic health in adolescents.

Physical activity and health research in adolescents has predominantly focused on the effect of endurance exercise, usually ≥30 min in duration and of moderate intensity, on insulin sensitivity [8,9]. Whilst such prolonged, moderate intensity, endurance exercise transiently improves metabolic health in adolescents (as shown by the 12–15% reduction in postprandial plasma insulin area under the curve [8,9]), it is not replicative of the activity patterns of young people and lacks ecological validity [10]. Adolescents typically undertake intermittent bouts of high-intensity activity, interspersed with short rest periods [10,11], similarly to the patterns observed in games-based activity. Yet, despite high-intensity intermittent activity being an ecologically valid mode of exercise in young people, there is limited research into its effect on adolescent metabolic health. Of the limited research conducted, high-intensity intermittent cycling [12] and 60 min of games-based activity in adolescents [13] has reduced plasma insulin incremental area under the curve (iAUC) by 24–30%. Therefore, intermittent activity elicits a greater response than the 12–15% reduction following more prolonged moderate intensity exercise [8,9], whilst also being deemed to have greater ecological validity for young people.

Whilst early findings suggested that intermittent exercise is an ecologically valid mode of physical activity that is effective in enhancing insulin sensitivity in adolescents, information relating to the optimum duration of intermittent activity for the enhancement of insulin sensitivity in children, adolescents and adults is unknown. Whilst the guidelines in the UK and Worldwide largely agree that 60 min physical activity per day is the recommended amount for young people [7], it is important to consider that young people typically do not perform this in a single bout [14]. Therefore, it is important to determine whether shorter durations of intermittent activity, that may be more attainable for young people, elicit similar protective metabolic effects, as seen following a 60-min bout of activity.

Furthermore, it is important to determine the residual effects of exercise on insulin sensitivity to inform exercise prescription (in particular exercise frequency) in adolescents. However, there is limited information available on the effect of intermittent activity on insulin sensitivity beyond 3 h post-exercise in adolescents [8,12,13]. When insulin sensitivity has been assessed up to 24 h post-exercise, the homeostatic model assessment (HOMA) has been the chosen marker of insulin resistance [8,12,13]. Following high-intensity intermittent cycling [8,12] and games-based activity [13], no effect was observed on HOMA when compared with the rested state in healthy adolescents. However, as HOMA is a fasted measure of hepatic insulin sensitivity, it is not sensitive to the changes in peripheral insulin sensitivity that occur post-exercise [12]. Therefore, the oral glucose tolerance test (OGTT) and the postprandial glycaemic and insulinaemic response to a standardised meal are potential alternatives that assess peripheral insulin sensitivity, with the latter having greater ecological validity. To date, only the OGTT has been utilised in research and reported a 13% improvement in insulin sensitivity 24 h post high-intensity intermittent cycling [12], when compared to a rested trial. 

Finally, throughout adolescence, differences in insulin sensitivity exist between the sexes, with the postprandial peak in insulin following a standardised mixed meal being 30–40% greater in adolescent girls (for peak plasma insulin concentrations) when compared with adolescent boys [15]. Yet, there has been no research to date to ascertain whether the sex differences in the postprandial insulin resistance persist following an acute bout of exercise and should exercise reduce the postprandial response, whether exercise can be used as a potential intervention to reduce insulin resistance in adolescent girls.

Therefore, the main aim of the present study was to examine the effects of differing durations (30 min vs. 60 min) of high-intensity intermittent activity on postprandial glycaemic and insulinaemic responses in adolescents, both 3 h and 24 h post-exercise. The present study also aimed to determine whether intermittent activity reduced the magnitude of the difference in insulin sensitivity between adolescent boys and girls.

## 2. Methodology 

### 2.1. Participant Characteristics

Forty-one participants (13.6 ± 0.5 year) were recruited to participate in the present study. However, based on exclusion criteria, two participants were removed from the study due to an inability to undertake the 60 min of high-intensity intermittent activity (*n* = 1) and the presence of a congenital heart condition (*n* = 1). Therefore, thirty-nine participants (16 boys and 23 girls) completed the study. During familiarisation, body mass (Seca 770 digital scale, Hamburg, Germany), stature and sitting stature (Leicester Height Measure, Seca, Hamburg, Germany) were measured and subsequently used to calculate age at peak height velocity [16]. For descriptive purposes, four skinfold sites (tricep, subscapular, supraspinale and front thigh) and waist circumference were measured. The participants’ anthropometric characteristics are displayed in Table 1. 

### 2.2. Study Design

Following institutional ethical approval (Nottingham Trent University Ethical Advisory Committee), written informed parental/guardian consent and participant assent were obtained. Parents/guardians also completed a health screen questionnaire on behalf of the participant to ensure there were no medical conditions affecting participation in the study.

During familiarisation, participants had the main experimental protocol explained to them and were allowed the opportunity to ask any questions they may have. In groups of eight, the participants then completed the multi-stage fitness test (MSFT) and peak oxygen consumption (44.6 ± 4.8 mL·kg^−1^·min^−1^) was predicted using an adolescent specific equation [17]. Participants then returned to the classroom and were familiarised with a capillary blood sample and a battery of cognitive function tests, the results of which are not included in this study. To ensure the participants were able to comply with the high-intensity intermittent activity, a practice 15-min block of the Loughborough Intermittent Shuttle Test (LIST) was performed (described in detail below).

### 2.3. Main Trials

Participants completed three main trials (30-min exercise, 60-min exercise and rested control) in a randomised, counterbalanced, crossover order (each separated by at least 7 day). Participants recorded a food diary 24-h prior to the first main trial and during the evening of day one of the study. Recorded diets were repeated for subsequent experimental trials. Participants refrained from physical activity 24-h prior to and during all experimental trials. Parents/guardians were contacted the evening before each main trial to ensure compliance with these requirements.

Participants arrived at school (~8.30 a.m.) on the day of each experimental trial, following an overnight fast (from 9 p.m. the previous evening). A heart rate monitor (Team Sports System, Firstbeat Technologies Ltd., Jyvaskyla, Finland) was worn throughout day one of each experimental trial. Participants consumed a standardised breakfast (cornflakes, milk, and toast with margarine) on day one and two, and a standardised lunch (chicken sandwich, baked salted crisps and an apple) on day one only [13,18]. Each meal contained 1.5 g carbohydrate per kg body mass. Participants had 15 min to consume the standardised meals and water was allowed ad libitum. See Figure 1 for schematic of protocol.

#### 2.3.1. Capillary Blood Samples

During day one of the main trials, capillary blood samples were taken at baseline, immediately post-exercise and 60-min post-exercise. Further capillary blood samples were taken 30-min, 60-min (2-h post-exercise) and 120-min (3-h post-exercise) following the standardised lunch. On day two of the main trials, a fasted capillary blood sample was taken. Following the consumption of the standardised breakfast, further blood samples were taken at 30 min, 60 min and 120 min. 

For all samples, concentrations of blood glucose and plasma insulin were determined in duplicate using commercially available kits (glucose: GOD/PAP method, GL364, Randox, Crumlin, Ireland; insulin: ELISA, Mercodia Ltd, Uppsala, Sweden). Blood glucose and plasma insulin tAUC following the standardised lunch on day one and the standardised breakfast on day two were calculated using previously described methods [19]. HOMA-IR was calculated as fasted insulin (mU·L) × fasting glucose (nmol·L)/22.5 [20].

#### 2.3.2. Exercise Protocol

During the exercise trials, participants completed either 30 min or 60 min of high-intensity intermittent activity, in the form of the Loughborough Intermittent Shuttle Test (LIST) [21]. During the LIST, participants ran between two markers, separated by 20 m, to pre-determined speeds dictated by an audio signal. The exercise pattern consisted of three 20-m shuttles at walking pace, a 15-m sprint followed by rest (8 s total duration), three 20-m shuttles at 85% VO_2_ peak and three 20-m shuttles at 55% of VO_2_ peak (percentage of VO_2_ peak determined from performance on the MSFT). Sprint times were recorded using infrared timing gates (Brower Timing Systems IRD-T173, Draper, UT, USA) and average sprint times for each block were calculated. The above pattern was repeated eight times, lasting ~12-min (as presented in Figure 2). The 30-min trial consisted of 2 blocks and the 60-min trial 4 blocks, with a 3-min recovery provided between blocks.

#### 2.3.3. Statistical Analysis

All data were analysed using SPSS (Version 24, SPSS Inc, Chicago, IL, USA). Data were assessed for normality using the Shapiro–Wilk test, which revealed that all dependent variables were normally distributed (all *p* > 0.05). Data were also assessed for homogeneity of variance using Mauchly’s test of Sphericity, which revealed that data was distributed equally. Blood glucose and plasma insulin concentration data were analysed via three-way (trial * time * sex) analysis of variance (ANOVA) with repeated measures for trial and time. Separate ANOVAs were conducted for day one and day two separately. Where significant interactions were observed post-hoc pairwise comparisons were performed via t-tests at each time point with Bonferroni correction. Blood glucose tAUC, plasma insulin tAUC, sprint times and heart rate were compared using two-way mixed method ANOVA, with sex as the between subjects factor and repeated measures for each trial. Where statistically significant differences existed effect sizes were calculated (Cohen’s d). For all analyses, significance was accepted as *p* < 0.05 and data are presented as mean ± SD.

## 3. Results

### 3.1. Performance Variables

Maximum heart rate was higher on the 60-min exercise trial (199 ± 8 beats·min^−1^) when compared to the 30-min exercise trial (195 ± 9 beats·min^−1^, t_(29)_ = −4.2, *p* < 0.001). Average sprint time during the 30-min LIST (block 1: 3.08 ± 0.29 s; block 2: 3.11 ± 0.29 s) and 60-min LIST (block 1: 3.08 ± 0.19 s; block 2: 3.13 ± 0.22 s; block 3: 3.13 ± 0.27 s; block 4: 3.16 ± 0.27 s) did not differ across each block of the LIST completed (main effect of trial, F_(5,135)_ = 2.50, *p* = 0.966). 

### 3.2. Glycaemic Response 

#### 3.2.1. Day One

Overall, blood glucose concentration on day one of the study did not differ between the 30-min LIST trial, 60-min LIST trial and the rested control trial (main effect of trial, *p* = 0.762), yet did change across time (main effect of time, F_(5,120)_ = 36.5, *p* < 0.001). The pattern of change in blood glucose concentration differed across trials (trial * time interaction, F_(6,168)_ = 3.6, *p* = 0.004; Figure 3); whereby blood glucose concentration was lower 1 h post-exercise during the 30-min LIST trial (30-min LIST: 3.6 ± 0.6 mmol·L^−1^) compared to the rested control trial (rested: 4.1 ± 0.9 mmol·L^−1^, F_(2,27)_ = 4.8, *p* = 0.001, d = 0.67). When considering the effect of sex, the glycaemic response did not differ between males and females (main effect of sex, *p* = 0.204), nor did the pattern of change in the glycaemic response differ between boys and girls (all *p* > 0.05; Appendix A). 

The tAUC for postprandial blood glucose concentration following the standardised lunch did not differ between trials (main effect of trial, *p* = 0.520). Yet, when considering the effect of sex there was a difference between boys and girls, whereby tAUC for postprandial blood glucose concentration was higher in boys than girls (boys: 560.2 ± 14.6 mmol·L^−1^ × 120 min; girls: 519.8 ± 11.6 mmol·L^−1^ × 120 min; main effect of sex, F_(1,37)_ = 4.7, *p* = 0.040). In contrast, there was no interaction effect between trial and sex (trial * sex interaction, *p* = 0.391; Appendix A). 

#### 3.2.2. Day Two

Overall, blood glucose concentration following the consumption of the standardised breakfast (day two) did not differ between trials (main effect of trial, *p* = 0.444), yet did differ across time (main effect of time, F_(3,105)_ = 86.0, *p* < 0.001). The pattern of change in postprandial blood glucose concentration did not differ between trials (trial*time interaction, *p* = 0.549). When considering the effect of sex, the glycaemic response did not differ between males and females (main effect of sex, *p* = 0.578), nor did the pattern of change in the glycaemic response differ between males and females (all *p* > 0.05). Finally, tAUC for blood glucose concentration following the standardised breakfast did not differ between trials (main effect of trial, *p* = 0.245), or sexes (main effect of sex, *p* = 0.278), nor was there an interaction between trial and sex (trial * sex interaction, *p* = 0.531; Appendix A). 

### 3.3. Insulinaemic Response 

#### 3.3.1. Day One

Overall, plasma insulin concentration on day one of the study did not differ between trials (main effect of trial, *p* = 0.091), yet changed across time (main effect of time, F_(5,165)_ = 66.5, *p* < 0.001). The pattern of change in plasma insulin concentration differed across trials (trial*time interaction, F_(10,330)_ = 3.2, *p* = 0.012, Figure 4), with reduced plasma insulin concentration 1 h post-exercise during the 60-min LIST trial when compared with the rested trial (60-min LIST: 58.3.0 ± 46.7 pmol·L^−1^, rested trial: 89.3 ± 82.2 pmol·L^−1^, t_(36)_ = 2.6, *p* = 0.011, d = 0.48). Peak postprandial plasma insulin concentration remained lower during the 60-min LIST trial when compared with the rested control (60-min LIST: 271.0 ± 151.4 pmol·L^−1^, rested trial: 344.7 ± 198.6 pmol·L^−1^, t_(38)_ = 3.1, *p* = 0.004, d = 0.42) and 1 h following the standardised lunch (60-min LIST: 195.6 ± 130.8 pmol·L^−1^, rested trial: 251.3 ± 173.3 pmol·L^−1^, t_(38)_ = −2.4, *p* = 0.011, d = 0.38). When considering the effect of sex, overall plasma insulin concentration was lower in boys when compared with girls (boys: 137.3 ± 15.3 pmol·L^−1^, girls: 184.3 ± 15.3 pmol·L^−1^, main effect of sex, F_(1,27)_ = 5.5, *p* = 0.031). Yet, the pattern of change in plasma insulin concentration did not differ between males and females (all *p* > 0.05; Appendix A). 

The tAUC for the insulinaemic response to the standardised lunch was lower on the 60-min LIST trial when compared to the rested control trial (60-min LIST: 21,754 ± 16,861 pmol·L^−1^ × 120 min, resting: 26,931 ± 21,634 pmol·L^−1^ × 120-min, main effect of trial, F_(2,74)_ = 4.3, *p* = 0.032, d = 0.44, Figure 5). Furthermore, there was a tendency for tAUC for the insulinaemic response to the standardised lunch to be lower on the 30-min LIST trial when compared to the rested control trial (30-min LIST: 24,273 ± 16,131 pmol·L^−1^ × 120 min, *p* = 0.080, d = 0.22). When considering the effect of sex, tAUC was lower in boys compared to girls (males: 20,445 ± 9556 pmol·L^−1^ × 120 min, females: 29,876 ± 15,748 pmol·L^−1^ × 120 min, main effect of sex, F_(1,28)_ = 7.4, *p* = 0.011, d = 0.75), yet the pattern of change between trials did not differ across the sexes (trial * sex interaction, *p* = 0.677; Appendix A). 

#### 3.3.2. Day Two

Overall, the plasma insulin concentration following the consumption of the standardised breakfast (day two) did not differ between trials (main effect of trial, *p* = 0.131), yet did change across time (main effect of time, F_(3,105)_ = 91.8, *p* < 0.001). The pattern of change in the insulinaemic response post-breakfast did not differ between the trials (trial*time interaction, *p* = 0.252). When considering the effect of sex, the postprandial insulinaemic response did not differ between boys and girls (main effect of sex, *p* = 0.089). Yet, the pattern of change in plasma insulin concentration across time differed between boys and girls (time*sex interaction, F_(3,105)_ = 4.0, *p* = 0.017), whereby girls had a higher plasma insulin concentration 30 min following the consumption of the standardised breakfast when compared with boys (girls: 530.9 ± 250.7, boys: 393.3 ± 195.6; F_(1,35)_ = 5.4, *p* = 0.012; Appendix A). 

Finally, the tAUC for the insulinaemic response to a standardised breakfast did not differ across trials (main effect of trial, *p* = 0.322) but was higher in girls compared to boys (girls: 36714 ± 30511 pmol·L^−1^ × 120-min, boys: 26202 ± 24688 pmol·L^−1^ × 120 min, main effect of sex, F_(1,37)_ = 7.3, *p* = 0.014, d = 0.38). The pattern of change in the tAUC insulinaemic response did not differ across trials between males and females (trial * sex interaction, *p* = 0.356; Appendix A). 

### 3.4. HOMA-IR

HOMA-IR was calculated for the fasted blood samples on day one and day two, with no overall difference between trials (main effect trial, *p* = 0.145), or between day one and day two (main effect time, *p* = 0.562). Furthermore, the pattern of change in HOMA-IR between day one and day two was similar between trials (trial*time interaction, *p* = 0.513). When considering the effect of sex, there was no difference in HOMA-IR between males and females (main effect of sex, *p* = 0.278), nor did the pattern of change in HOMA-IR differ between the sexes (all *p* > 0.05).

## 4. Discussion

The present study is the first to examine the glycaemic and insulinaemic responses to high-intensity intermittent exercise (HIIE) of different durations, for up to 24 h post-exercise, whilst considering the potential moderating effect of sex on these responses during adolescence. The main finding of the study was that the postprandial insulinaemic response to a standardised lunch was lower following 60-min HIIE and tended to be lower following 30-min HIIE when compared to the rested control trial. Furthermore, blood glucose concentration was lower 1-h post-exercise on the 30-min trial when compared to the resting trial. However, postprandial glycaemic and insulinaemic responses to a standardised breakfast 24-h post-exercise were not affected. Furthermore, when considering the effect of sex on insulin sensitivity, girls had consistently higher plasma insulin concentrations in comparison to boys, yet the post-exercise responses were not affected by sex.

In the present study plasma insulin concentration decreased by 35% 1 h following 60 min of HIIE when compared to a rested control trial. These beneficial effects remained following the consumption of a standardised meal, with a 21% decrease in peak postprandial plasma insulin concentration. Postprandial plasma insulin tAUC on day one was also lower during the 60-min LIST trial (−20%) and 30-min LIST trial (−11%) when compared to the rested control. The reduction in postprandial plasma insulin concentration following HIIE is indicative of enhanced insulin sensitivity in healthy adolescents, with less insulin required to regulate blood glucose concentration. Whilst post-exercise enhanced insulin sensitivity has previously been documented in adolescents [8,13], the novel findings of the present study suggest that both 30 and 60 min of high-intensity intermittent running are sufficient to enhance insulin sensitivity on the day the exercise was undertaken; and this is the first study to compare exercise of differing durations. The findings suggest that exercise duration mediates the magnitude of the postprandial insulinaemic response, with greater effects seen following 60- compared to 30-min HIIE.

The 12% reduction in blood glucose concentration observed 1-h post 30-min HIIE is consistent with previous findings, whereby blood glucose concentration was reduced by 11% following 60-min of games-based activity [13] and ~15% post 45 min of moderate intensity activity [9]. However, the findings of the present study are novel as the reduction in blood glucose concentration occurred during the 30-min LIST trial, the duration of which is significantly less than has previously been observed in adolescents [9,13] and is shorter than the government guidelines of 60-min moderate-to-vigorous physical activity per day. These findings are promising, as the majority of adolescents in the UK do not currently meet the recommended physical activity guidelines and those that do, do so through shorter accumulated bouts [14]. The findings of the present study have important practical implications, as shorter bouts of HIIE can be recommended as an effective and ecologically valid mode and duration of exercise to reduce blood glucose concentration in adolescents. The findings also in agreement with training studies that report HIIE reduces the presence of metabolic biomarkers in young people [22], whilst including important details relating to exercise duration and can therefore be used to inform future therapeutic interventions that are achievable for young people to implement into their daily lives and to reverse the adverse cardiometabolic health trends currently observed in young people [23]. Future work should continue to assess the effect of exercise duration on blood glucose concentration to determine whether accumulative, shorter bouts (<30-min) of exercise also reduce postprandial blood glucose concentration. 

Despite the reduction in blood glucose concentration 1-h post-exercise, there were no differences in the postprandial glycaemic response to a standardised meal (tAUC) between trials. These findings are consistent with those of Dring et al. [13], whereby 60-min of games-based activity did not reduce the glycaemic response to a standardised meal in comparison to a rested trial. Whilst these corroborating findings suggest intermittent activity in adolescents does not affect postprandial blood glucose concentration, high-intensity intermittent cycling (8 × 1 min cycling at 90% peak power) has previously reduced the glycaemic response (8% reduction in tAUC) following an OGTT in adolescent boys [8]. Such discrepancies might be explained in part by the different characteristics of the participants assessed. Predicted VO_2_ peak is a potential moderating variable, as the VO_2_ peak of the participants in the present study is on the 75th percentile for adolescents of this age [24], whereas the VO_2_ peak of participants from previous studies has been ≤50th percentile for their chronological age [8]. Participants with a greater predicted VO_2_ peak (from performance on the MSFT) might require more intense exercise stimuli (through increased intensity or duration) to reduce the postprandial glycaemic response following exercise, whereas lower fit adolescents may respond positively to a lower exercise stimulus [24]. 

The present study also examined the residual glycaemic and insulinaemic responses to a standardised breakfast 24-h following exercise. Interestingly, there was no difference in the postprandial glycaemic and insulinaemic responses to the standardised breakfast across trials in the present study. The present study is the first to assess the postprandial glycaemic and insulinaemic responses to an ecologically valid meal up to 24-h post-exercise, and suggests that daily physical activity is required to optimise cardiometabolic health in young people. 

Finally, in the present study plasma insulin concentrations were consistently higher in girls than boys, yet the response to exercise did not differ between the sexes. The finding that girls exhibit greater insulin resistance during adolescence is consistent with previous research whereby the insulinaemic response to a mixed meal was 30–40% higher in girls than boys [15]. The present study aimed to examine whether HIIE could reduce the postprandial differences in insulin sensitivity observed between the sexes and attenuate the insulin resistance observed in adolescent girls. However, on the day exercise was undertaken and the day following, the insulin response was consistently elevated in girls compared with boys; an effect not influenced by HIIE. Future studies should continue to assess whether different modes, durations and intensities of exercise are able to attenuate the difference in insulin sensitivity observed between the sexes during adolescence, given the importance for cardiometabolic health. 

## 5. Conclusions

Overall, the findings of the present study suggest that both 30- and 60-min high-intensity intermittent running are ecologically valid forms of exercise that enhance the regulation of blood glucose and insulin sensitivity in healthy adolescent boys and girls. However, the magnitude of the response for insulin sensitivity is greater following 60 min of high-intensity intermittent running than 30-min, when exercise intensity is matched across trials. In addition, the effect of the exercise on blood glucose and plasma insulin was only maintained on the day the exercise was performed, with no effect on the glycaemic and insulinaemic response to a standardised breakfast 24 h post-exercise. Therefore, the findings of the present study support the government physical activity guidelines that suggest young people should participate in daily moderate-to-vigorous physical activity and promote high-intensity intermittent running between 30–60 min in duration as a potential therapeutic intervention to protect against the development of type 2 diabetes in adolescents. 

## Figures and Tables

**Figure 1 nutrients-12-00754-f001:**
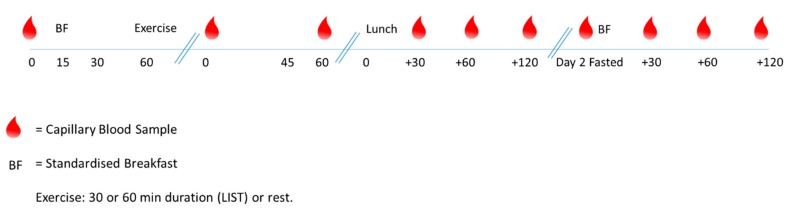
Schematic of protocol.

**Figure 2 nutrients-12-00754-f002:**
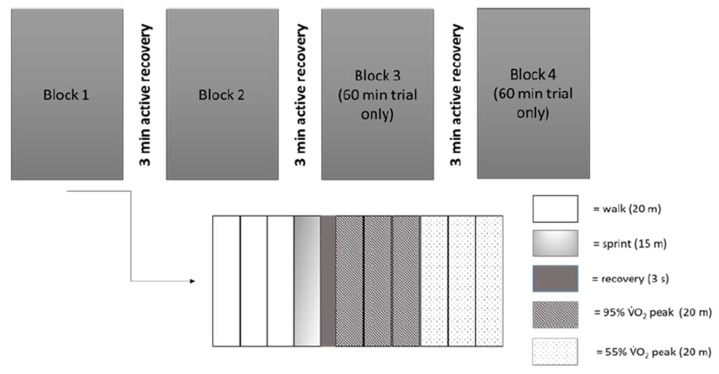
Overview of the Loughborough Intermittent Shuttle Test used during the 30-min and 60-min exercise trial.

**Figure 3 nutrients-12-00754-f003:**
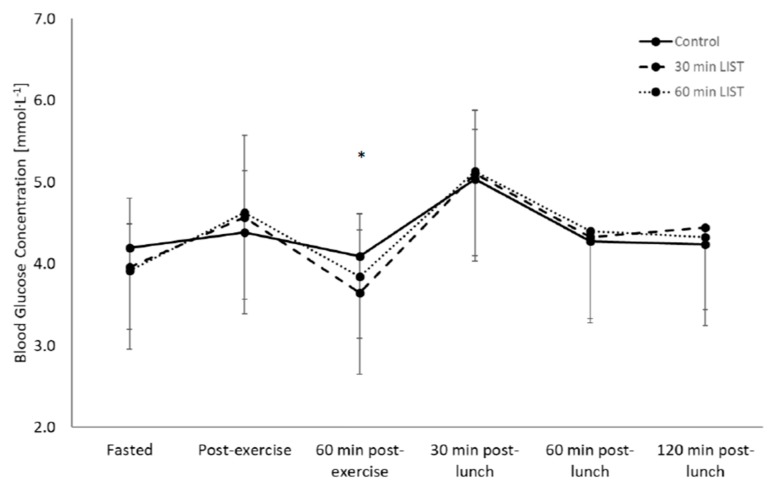
Glycaemic response during the 30-min LIST trial, 60-min LIST trial and rested control trial on day one of the study (mean ± SD), trial * time interaction, F_(6,168)_ = 3.6, *p* = 0.004; * 30-min LIST trial significantly lower than the rested control trial, *p* = 0.001.

**Figure 4 nutrients-12-00754-f004:**
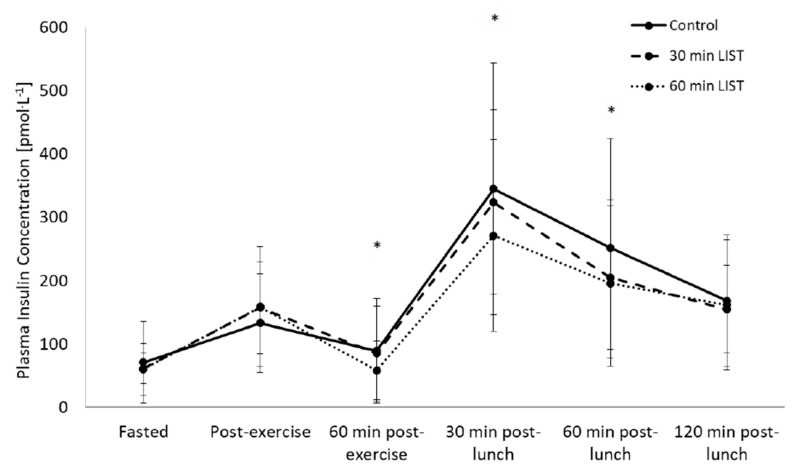
Insulinaemic response during the 30-min LIST trial, 60-min LIST trial and the rested control trial on day one of the study (Mean ± SD), trial*time interaction, F_(10,330)_ = 3.2, *p* = 0.012, * 60-min LIST trial significantly lower than the rested control trial, *p* = 0.011).

**Figure 5 nutrients-12-00754-f005:**
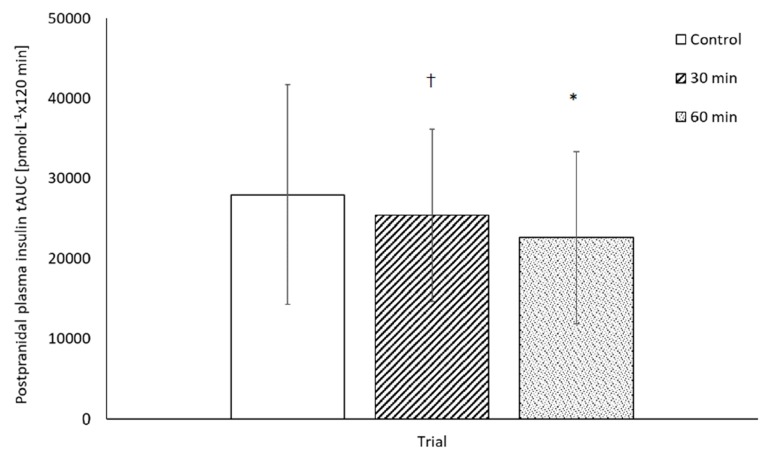
Plasma insulin total area under the curve following consumption of a standardised lunch on the 30-min LIST trial, 60-min LIST trial and rested control trial (mean ± SD), main effect of trial, F = 4.3, *p* = 0.030, * 60-min LIST trial < rested control trial, *p* = 0.032; ^†^ tendency for 30-min LIST trial < rested control trial, *p* = 0.080).

**Table 1 nutrients-12-00754-t001:** Anthropometric characteristics.

	Overall (*n* = 39)	Boys (*n* = 16)	Girls (*n* = 23)	*p* Value ^a^
Age (year)	12.4 ± 0.4	12.0 ± 0.3	12.3 ± 0.4	0.510
Height (cm)	157.8 ± 7.5	157.0 ± 9.9	159.3 ± 5.5	0.281
Body Mass (kg)	45.1 ± 7.1	43.7 ± 7.9	46.3 ± 7.2	0.660
Maturity Offset (year) ^b^	−1.4 ± 0.6	−1.7 ± 0.4	−1.2 ± 0.6	0.359
Waist Circumference (cm)	65.0 ± 4.8	64.6 ± 4.1	65.4 ± 5.4	0.202
Sum of Skinfolds (mm)	45.0 ± 14.0	41.6 ± 11.3	65.4 ± 5.4	0.605

^a^ comparison between boys and girls. ^b^ calculated using the method of Moore et al. [16].

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
