# Peer review of "Effect of Exercise Duration on Postprandial Glycaemic and Insulinaemic Responses in Adolescents"

_nutrients, 2020, doi:10.3390/nu12030754_

Round 1

Reviewer 1 Report

Here, the authors examine the effect of a 30 or 60 minute high intensity intermittent exercise (HIIE) paradigm (LIST) on the postprandial glucose and insulin responses in adolescent male and female participants.  The authors show that plasma glucose concentration is significantly lower one hour after 30 minutes of HIIE compared to control (no exercise).  They also demonstrate that postprandial insulin levels are significantly lower following 60 minutes of HIIE, with a trend toward a decrease in postprandial insulin following 30 minutes of HIIE.  

The overall purpose of this research is to determine the duration of a high intensity interval exercise paradigm for healthy adolescents that will ultimately prevent against the development of type II diabetes.  The authors provide a small piece of the puzzle, investigating the effect of a single bout of HIIE on glucose and insulin concentrations post exercise (and following a meal). 

Overall, the experimental design is sound and the manuscript is well-written.  This reviewer has a few comments and several minor edits:

Comments: 

Line 135: It is unclear exactly when the exercise was performed.  The authors should consider including a figure illustrating their experimental timeline including times and duration of exercise, meals, and blood sample collections. This would help to clarify the timing of blood samples outlined in lines 135-9. 

Statistical analyses: did the authors test for homogeneity of variances and ensure the data pass this assumption before using ANOVAs?  They should do this and report that it has been done.

The authors indicate that they perform multiple post hoc t-tests, but conducting multiple t-tests increases the risk of a type I error.  The authors should consider performing more stringent post-hoc analysis. 

The authors should try to relate their findings to what is known about the long-term effects of HIIE on type II diabetes in their discussion.   

Minor edits:

Line 17: tAUC should be defined

Line 19 and 21: The authors should say when compared with the ‘resting condition’, instead of when compared with ‘resting’. 

Line 68: the authors indicate that there is little known about the effect of the frequency of exercise on insulin sensitivity, but they do not address the frequency of exercise in this study (unless they are referring to the frequency of the exercises within the HIIE).  This reviewer doesn’t see how this statement fits into that paragraph or the overall paper.    

The authors referring to HIIE as an ‘ecologically valid mode of physical activity’ gets repetitive in the introduction. 

Table 1: In the left column, it says ‘Boddy mass’ instead of ‘Body mass’

Line 108: The name of the specific committee from which they received ethical approval should be included. 

Line 117: The authors should say that the battery of cognitive function tests are ‘not included in this study’ instead of ‘will be published elsewhere’

Line 118: The authors should indicate that LIST will be described below.

Line 174: This reviewer is questioning the heart rate data that the authors suggest is highly significant, but the means are 199 +/- 8 bpm vs 196 +/- 9 bpm.  Are these really significantly different?  Can the authors look into this? 

Figure 3: There two * on top of one another in Figure 3 above the 60 min post-exercise data

Author Response

The authors would like to thank the reviewer for taking the time to read and provide useful comments on our manuscript, which we believe has led to an overall improvement of the paper, it is greatly appreciated. We have responded to each of the comments below and have made changes in the paper in red font for ease of access. 

Comments: 

Line 135: It is unclear exactly when the exercise was performed.  The authors should consider including a figure illustrating their experimental timeline including times and duration of exercise, meals, and blood sample collections. This would help to clarify the timing of blood samples outlined in lines 135-9. 

Thank you for highlighting that this information was missing and was therefore affecting the clarity of our description. We have added a schematic for the protocol, with a timeline for each of the blood samples and when the exercise was undertaken (we refer to this in the methods section). We hope that this has improved the methodology (line 135). 

Statistical analyses: did the authors test for homogeneity of variances and ensure the data pass this assumption before using ANOVAs?  They should do this and report that it has been done.

The homogeneity of variance was previously assessed, however we did not comment on this in the statistical analysis. We can confirm that using Mauchly's test of Sphericity, data were equally distributed with sphericity assumed and the apporpriate statistical action taken thereafter. Comment has been made upon this in the text on lines 166-168. 

The authors indicate that they perform multiple post hoc t-tests, but conducting multiple t-tests increases the risk of a type I error.  The authors should consider performing more stringent post-hoc analysis. 

Thank you for the above comment - we realise that there was an oversight as we did not state that for the mulitple post-hoc tests Bonferroni corrections had been applied We have ensure that this is now stated in the statistical analysis on lines 171 - 172.

The authors should try to relate their findings to what is known about the long-term effects of HIIE on type II diabetes in their discussion.   

 Again, thank you for the above comment, we agree that it is important to relate these findings to the training studies that have been conducted to date on HIIE. To reference these studies, the authors have cited a systematic review of HIIE training studies, that have examined the effect of exercise on cardiometabolic risk factors in young people. We believe the studies included in the review support our findings and promote exercise as a therapeutic intervention, whilst detailing the finer details required for exercise prescription (lines 307 - 312).

Minor edits:

Minor edits, inlcuding definitions and grammatical errors have been corrected throughout the manuscript following your helpful comments. Thank you for detailing these as these changes were essential in improving the manuscript.

Line 68: the authors indicate that there is little known about the effect of the frequency of exercise on insulin sensitivity, but they do not address the frequency of exercise in this study (unless they are referring to the frequency of the exercises within the HIIE).  This reviewer doesn’t see how this statement fits into that paragraph or the overall paper.    

We thank you for this comment and would like to clarify what we originally intended this message to be as a research group. The authors were interested in establishing the duration in which the enhanced insulin state persisted following the exercise trials and whether this remained up to 24 h post-exercise. For us, this residual response will help in determining how frequently exercise needs to be undertaken to maintain insulin sensitivity (details that are important for exercise prescription).  

To help with this understanding we have reworded the opening sentence to the exercise frequency paragraph, so that it discusses the residual effect of exercise, with less of an emphasis on exercise frequency (line 68).

The authors referring to HIIE as an ‘ecologically valid mode of physical activity’ gets repetitive in the introduction.

Throughout the introduction we have reduced the number of times we reiterate HIIE being an ecologically valid mode of phyiscal activity. 

Line 174: This reviewer is questioning the heart rate data that the authors suggest is highly significant, but the means are 199 +/- 8 bpm vs 196 +/- 9 bpm.  Are these really significantly different?  Can the authors look into this? 

We would like to confirm that there was a typing error in the means of the heart rate (196 should have been 195) and that the statistical analysis was as reported in the original manuscript. Again, we would like to thank you for taking the time to carefully read the manuscript and highlighting key corrections. 

Reviewer 2 Report

In a randomised crossover 
study, the authors investigate the effects of high-intensity intermittent activity (at 30-min and 60-min)  on postprandial glycaemic and 
insulinaemic responses in adolescents boys and girls.

They found that the postprandial insulinaemic response to a standardised lunch was 
lower following 60-min high-intensity intermittent activity and tended to be lower following 30-min when compared to the rested control trial. Blood glucose concentration was lower 1-h post-exercise on the 30-min trial when compared to the resting trial. Postprandial glycaemic and insulinaemic responses to a standardised breakfast 24-h post-exercise were not affected. 

This is an interesting study, the manuscript is well-written, aims are clear and concise, data analysis are appropriate, interpretation of results, and conclusions as well.

I have no questions about the manuscript.

Author Response

The authors would like to thank the reviewer for taking the time to read our manuscript and provide the positive comments below, we greatly appreciate them. 

Thanks again.